# Galaxy clusters enveloped by vast volumes of relativistic electrons

V. Cuciti[1,2 ✉], F. de Gasperin[1,2], M. Brüggen[1], F. Vazza[2,3], G. Brunetti[2], T. W. Shimwell[4],
H. W. Edler[1], R. J. van Weeren[5], A. Botteon[2,3,5], R. Cassano[2], G. Di Gennaro[1], F. Gastaldello[6],
A. Drabent[7], H. J. A. Röttgering[5] & C. Tasse[8,9]

The central regions of galaxy clusters are permeated by magnetic fields and filled with relativistic electrons[1]. When clusters merge, the magnetic fields are amplified and relativistic electrons are re-accelerated by turbulence in the intracluster medium[2,3]. These electrons reach energies of 1–10 GeV and, in the presence of magnetic fields, produce diffuse radio halos[4] that typically cover an area of around 1 Mpc[2]. Here we report observations of four clusters whose radio halos are embedded in much more extended, diffuse radio emission, filling a volume 30 times larger than that of radio halos. The emissivity in these larger features is about 20 times lower than the emissivity in radio halos. We conclude that relativistic electrons and magnetic fields extend far beyond radio halos, and that the physical conditions in the outer regions of the clusters are quite different from those in the radio halos.

We used the LOw Frequency ARray (LOFAR[5]) to search for signatures of non-thermal radiation in the outer regions of galaxy clusters. We examined the radio emission at 144 MHz from 310 massive clusters from the Planck Sunyaev–Zel'dovich catalogue[6] in the LOFAR Two Meter Sky Survey (LoTSS[7,8]). After a careful removal of contaminating emission from other astronomical sources (Methods) we found that in four cases the radio halo emission is embedded in a much larger emission that extends over 2–3 Mpc and fills the volume of the cluster, at least up to $R_{500}$ (Fig. 1), that is the radius within which the mean mass over-density of the cluster is 500 times the cosmic critical density at the cluster redshift ($z$). The mass enclosed in a sphere with radius $R_{500}$ is $M_{500}$. These clusters are ZwCl 0634.1+4750 ($z = 0.17$, $M_{500} = 6.65 \times 10^{14} M_\odot$), Abell 665 ($z = 0.18$, $M_{500} = 8.86 \times 10^{14} M_\odot$), Abell 697 ($z = 0.28$, $M_{500} = 10.99 \times 10^{14} M_\odot$) and Abell 2218 ($z = 0.17$, $M_{500} = 5.58 \times 10^{14} M_\odot$). All these clusters are dynamically disturbed[9] and were known to host radio halos[10–12], yet the larger-scale emission discovered with LOFAR, that we call the 'megahalo', enables us to probe the magnetized plasma in a volume that is almost 30 times larger than the volume occupied by the radio halos.

The radial surface brightness profile of ZwCl 0634.1+4750 (Fig. 2) clearly demonstrates the difference to the radio halo emission. The profiles of the other three clusters are shown in Extended Data Fig. 3. All the profiles show two components: a bright region dominated by the radio halo, the brightness of which decreases relatively fast with cluster-centric distance and an extended, low-surface-brightness component. The region of the profile dominated by the radio halo can be fitted with an exponential function as commonly found for these type of sources[13]. The emission beyond the radio halo shows a shallower profile implying that at 600–800 kpc from the centre a transition occurs. The surface brightness of the large-scale emission is a factor of at least 10 lower than the surface brightness of the radio halo with an average emissivity of approximatly 20 to 25 times lower than the emissivity

of radio halos (Methods). The low surface brightness, combined with its large size, is the reason why this emission has eluded all previous searches but could be detected by LOFAR.

For two clusters, ZwCl 0634.1+4750 and Abell 665, we present deep observations at even lower frequencies (53 MHz and 44 MHz, respectively) for which low-energy relativistic electrons shine brightly. In the observation of ZwCl 0634.1+4750 only the brightest part of the large-scale emission beyond the radio halo is detected, whereas in Abell 665 almost all of it is visible (Extended Data Fig. 1). The combination of the low (144 MHz) and ultra-low (around 50 MHz) frequency observations enables us to constrain the energetics of the particles responsible for the synchrotron emission via the radio spectral index $\alpha$ (defined as $S(\nu) \propto \nu^\alpha$, with $S$ being the flux density and $\nu$ the frequency). We obtained $\alpha = -1.62 \pm 0.22$ and $\alpha = -1.64 \pm 0.13$ for ZwCl 0634.1+4750 and Abell 665, respectively. Although the uncertainties on the spectral index measurements are relatively large, in both cases we found evidence that the spectrum is steeper than the spectrum of the central radio halos, which is $\alpha = -1.25 \pm 0.15$ for ZwCl 0634.1+4750 and $\alpha = -1.39 \pm 0.12$ for Abell 665. This adds to the evidence that the megahalos are a phenomenon distinct from radio halos.

Our results confirm that magnetic fields and relativistic electrons fill a much larger volume than previously observed, therefore requiring ubiquitous mechanisms for the energization of particles on large scales. The existence of megahalos demonstrates that beyond the edge of radio halos mechanisms operate that maintain a sea of relativistic electrons at energies high enough to emit at frequencies of approximately 100 MHz.

The surface brightness of this feature stays fairly constant over more than 500 kpc, whereas the underlying intracluster medium (ICM) density decreases by a factor of around 5 (ref. [14]). This can be used to infer the relative contribution to the cluster energy content from thermal

[1]Hamburger Sternwarte, University of Hamburg, Hamburg, Germany. [2]INAF – Istituto di Radioastronomia, Bologna, Italy. [3]Physics & Astronomy Department, University of Bologna, Bologna, Italy. [4]ASTRON, The Netherlands Institute for Radio Astronomy, Dwingeloo, the Netherlands. [5]Leiden Observatory, Leiden University, Leiden, the Netherlands. [6]INAF – IASF Milano, Milano, Italy. [7]Thüringer Landessternwarte, Tautenburg, Germany. [8]GEPI & USN, Observatoire de Paris, Université PSL, CNRS, Meudon, France. [9]Department of Physics & Electronics, Rhodes University, Grahamstown, South Africa. ✉e-mail: vcuciti@hs.uni-hamburg.de

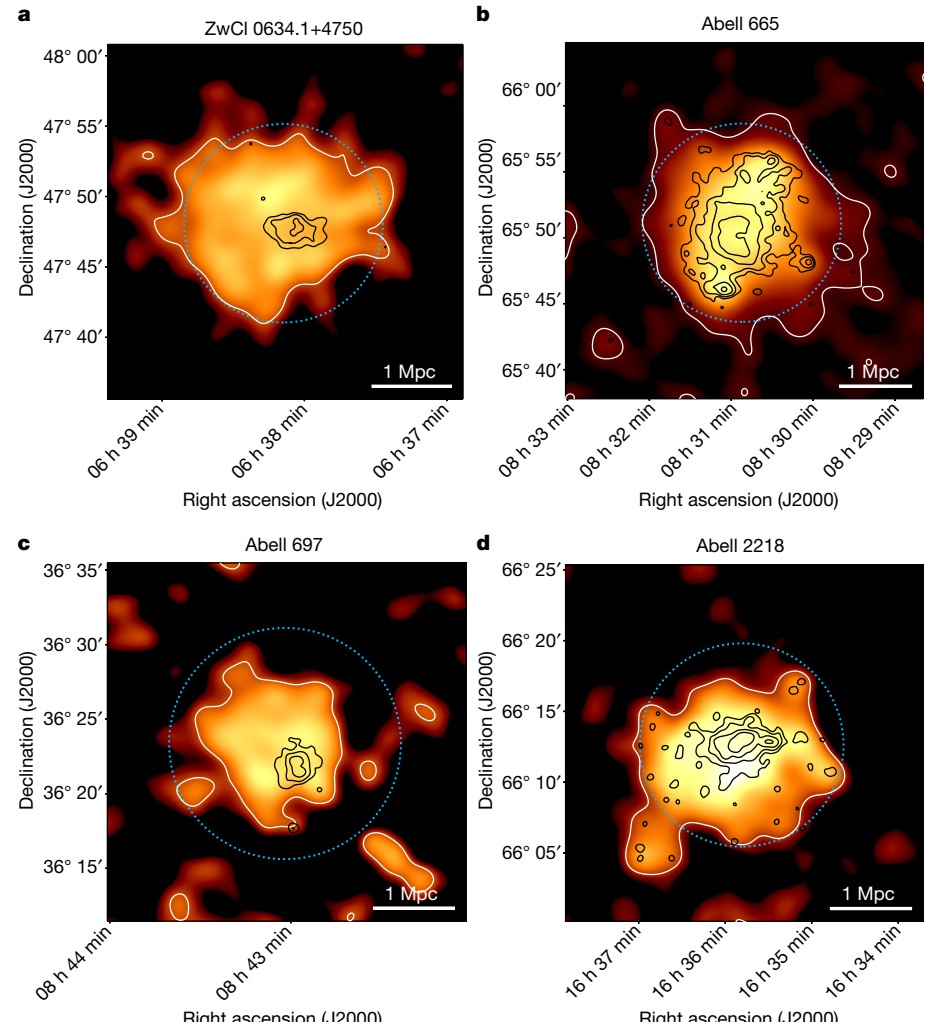

**a** ZwCl 0634.1+4750

**b** Abell 665

**c** Abell 697

**d** Abell 2218

**Fig. 1 | Radio images of the four galaxy clusters. a**, ZwCl 0634.1+4750. **b**, Abell 665. **c**, Abell 697. **d**, Abell 2218. Black contours: LOFAR 144 MHz at 30″ resolution, showing the location of the radio halos. Only sources in the high-resolution image have been subtracted. Contours at (4, 8, 16, 32) × σ. Orange image and white contours: LOFAR 144 MHz image at 2′ resolution. All sources, including the radio halos, have been subtracted. Contour at 3σ confidence level. The radius of the light blue circle corresponds to $R_{500}$.

and non-thermal components, which has implications, for example, on the Sunyaev–Zel'dovich effect in cosmology[15]. If the magnetic field scales with the ICM density as found in ref. [16] ($B^2 \propto n_{ICM}$, where $n_{ICM}$ is the density of the ICM), the ratio between the energy density of non-thermal electrons and the thermal gas energy must increase going towards the outer regions of the cluster. For example, for the cluster Abell 2218, whose ICM density profile has been studied in ref. [17], we found that a constant energy density of relativistic electrons with radius can reproduce the observed surface brightness profile. In this case, the ratio between the energy density of non-thermal electrons and the thermal gas energy must increase by a factor of around 3 going from approximately $0.5 \times R_{500}$ to approximately $R_{500}$. Alternatively, the magnetic field strength must approximately be increased by a factor of $\sqrt{3}$ over those distances to produce the same radio emission. Reproducing the observed trend of cosmic rays and magnetic fields in such peripheral regions of galaxy clusters, where a mixture of accretion modalities and (re)acceleration mechanisms is present, represents an important challenge for future theoretical models of galaxy clusters.

The steep spectrum that we observe in ZwCl 0634.1+4750 and Abell 665 indicates that turbulence might be responsible for maintaining the relativistic electrons inside a volume of the order of 10 Mpc³ (refs. [18–20]) and that we may be probing a turbulent component different from that powering radio halos. Numerical simulations seem to support

this scenario and show that, in addition to the central merger-driven turbulence responsible for radio halos, there is a broader turbulent component, probably related to the accretion of matter onto the cluster, that can accelerate particles[21–23] (Methods). The observed characteristics of megahalos suggest a change either in the macrophysical or in the microphysical properties of the plasma when moving from the radio halo to the outer region. In the former case, the properties of turbulence may change going towards the periphery, as suggested by simulations. In the latter case, microphysical properties such as the acceleration efficiency, the mean free path or transport properties, all of which are related, may change in the outer regions.

Although the mechanisms responsible for the formation of the large-scale emission are still unknown, it is reasonable to assume that the mass of clusters plays an important role in determining the energy budget available for particle acceleration, similar to what happens for radio halos. In fact, more powerful radio halos are hosted in more massive clusters[24]. To understand why we have detected this emission only and exactly in these four clusters, in Fig. 3 we show the mass–redshift distribution of the clusters inspected for this work. The three solid lines show the expected mass–redshift relation taking into account the cosmological surface brightness dimming (SB $\propto (1+z)^{-4}$) and assuming a power-law dependence of the large-scale emission surface brightness with the mass of the clusters (SB $\propto M^\beta$, where we assumed three

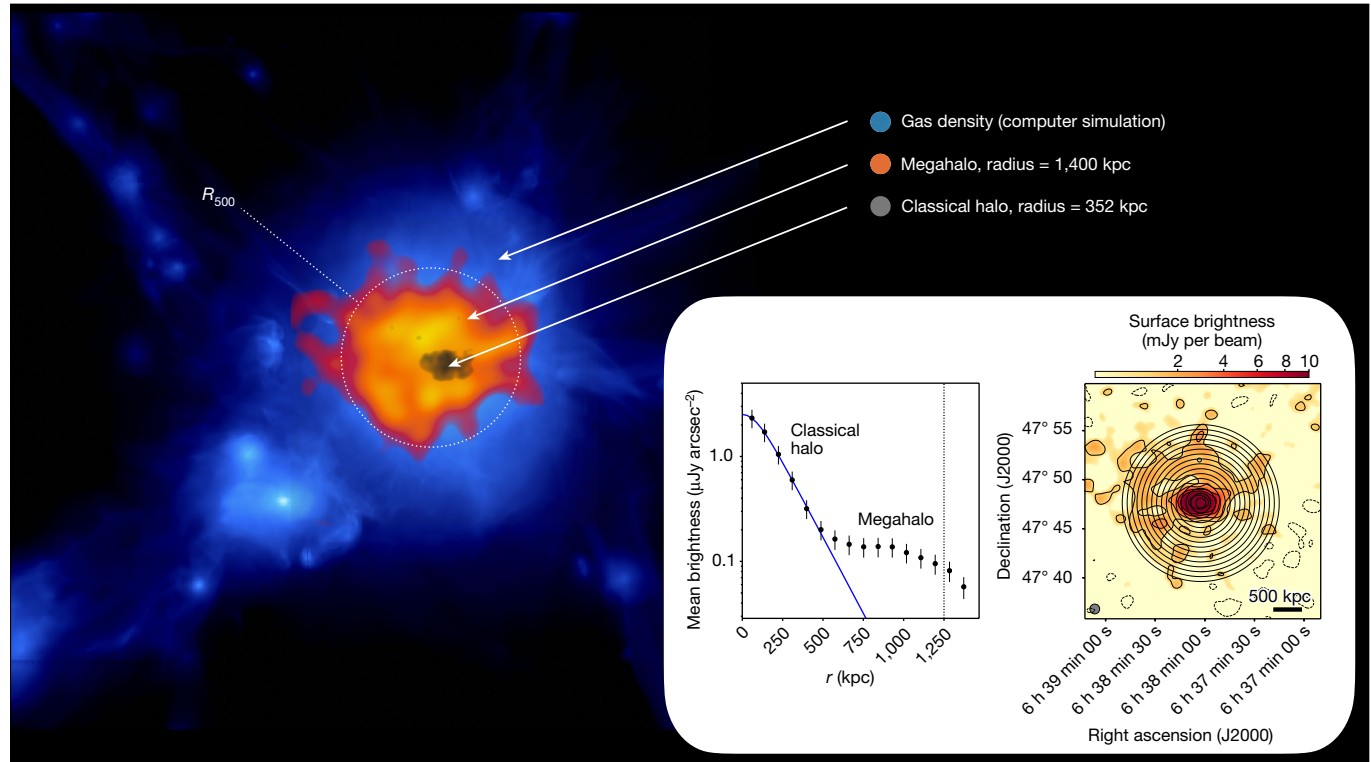

**Fig. 2 | Large-scale emission from the ICM in ZwCl 0634.1+4750.** In blue, the gas density distribution in a simulated massive galaxy cluster is shown[22] for illustrative purposes only. In orange, the LOFAR 144 MHz 2′ resolution image of the cluster ZwCl 0634.1+4750 is shown after the subtraction of all the sources in the field, including the radio halo. The central region, outlined in grey, shows the LOFAR 144 MHz 30″ resolution image of the classical radio halo. Inset: left, surface brightness radial profile of the diffuse radio emission in ZwCl 0634.1+4750. Error bars represent the 1σ uncertainty; right, the 60″ resolution 144 MHz image and the annuli we used to measure the surface brightness reported in the plot on the left (see Methods for more details).

possibilities for $\beta$). As the large-scale emission in ZwCl 0634.1+4750 is detected at around 3σ, we impose the condition that the lines go through that point in the diagram to set the normalization. This means that with current LOFAR observations we expect to be able to detect at more than 3σ the large-scale emission that embeds radio halos in clusters that lie above (that is, higher mass) and to the left (that is, lower $z$) of the assumed dependencies. Interestingly, all the sources that we detected lie in this region. The other clusters with similar masses and redshift either have low-quality data (because of observations being taken during disturbed ionospheric conditions that distort the radio signal at low frequencies) or they are in particularly complex fields for which an accurate subtraction of contaminating sources is not reliable[25,26]. The fact that the clusters where we discovered the new large-scale emission lie close or above our estimated detection limit in mass–redshift space suggests that we may be seeing just the tip of the iceberg of a phenomenon common to a large number of clusters that can come to light in deeper low-frequency radio observations.

In this Article, we have highlighted the differences between the diffuse emission in the central and outer regions of galaxy clusters. The results of this work may shed light on recent findings for a few clusters where it has been shown that radio halos become larger at lower frequencies, reaching largest linear sizes of the order of 2 Mpc (refs. [26–28]). Interestingly, all these clusters are massive and would lie on the top left region of the diagram in Fig. 3, supporting the idea of a new, emerging population of sources. Finally, evidence for a two-component nature of diffuse sources has been claimed in a few non-merging clusters[29–31]. However, they do not probe regions of galaxy clusters different from those of radio halos.

Currently, we can only observe megahalos in clusters that meet a certain combination of mass and redshift. However, our study suggests that deeper observations, such as those that will be made with the upgraded

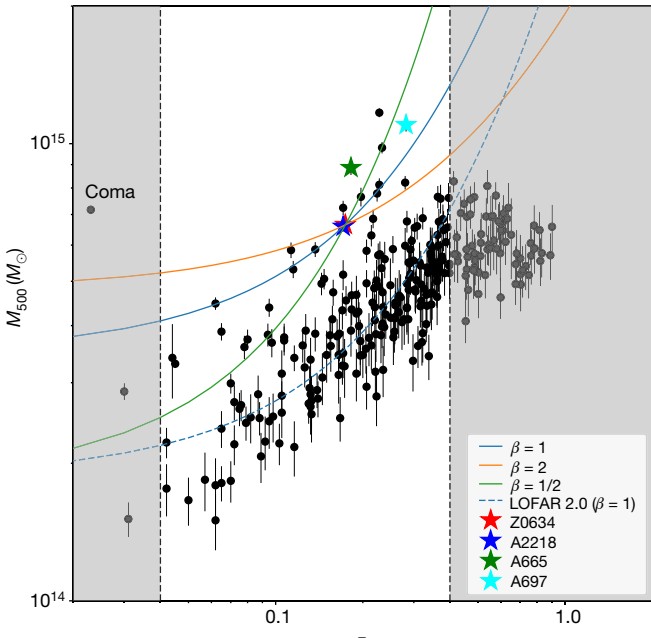

**Fig. 3 | Mass–redshift diagram for the Planck clusters in LoTSS.** Stars mark clusters where we found the megahalos. Solid lines define the region of the plot where we expect to be able to detect emission beyond radio halos with current LOFAR observations. Possible sources lying in the grey shaded regions would be resolved out (left) or unresolved (right). The dashed line marks the regions where we expect to be able to detect megahalos with LOFAR 2.0 ultra-low-frequency observations (the 1σ root-mean-square (r.m.s.) noise for sources with spectral index −1.6 is expected to decrease by a factor of 2 with respect to current observations). Error bars represent the 1σ uncertainty on the mass estimate[6].

LOFAR 2.0 and Square Kilometre Array[32], will unveil many more clusters showing diffuse emission on such large scales (Fig. 3), thus opening the possibility for systematic exploration of the peripheral regions of galaxy clusters. Whether megahalos constitute a new class of sources sitting below or embedding radio halos remains to be seen following deeper searches for this emission on larger samples. Whatever the case, the brightness and spectral index profiles suggest that a new type of phenomenology is at play when going to larger distances from the cluster centre.

These results show that relativistic electrons and magnetic fields fill larges swathes of the cosmos. It helps us to understand how energy is dissipated during the formation of large-scale structures as well as how particles are accelerated in low-density plasmas.

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

# Methods

## Observations and data reduction

The properties of the final images presented in this paper are listed in Extended Data Table 1, together with the source subtraction performed (see below) and the number of the figure in which they are presented.

## LOFAR high-band antenna data reduction

The LOFAR 144 MHz data presented in this paper are part of the LoTSS[7,8,33], which is an on-going 120–168 MHz survey of the entire northern hemisphere performed with the LOFAR high-band antennas (HBA). Data have been processed with the Surveys Key Science Project reduction pipeline v.2.2 (refs. [7,34]), which includes corrections for both direction-independent and direction-dependent effects (prefactor[35–37], killMS[38,39] and DDFacet[40]). We subtract all the sources outside a region of around 0.5 deg$^2$ containing the galaxy clusters from the $uv$-data by using the model produced by the pipeline. We then phase shift the dataset to the centre of the extracted region, we correct for the LOFAR station beam in that direction and we perform additional loops of phase and amplitude calibrations on the target field to improve the quality of the final image. This extraction procedure is presented in ref. [41] and is routinely used in LOFAR 144 MHz observations. The data reduction of the clusters of the Planck sample is discussed in detail in ref. [42]. Here we performed a more accurate procedure for the source subtraction aimed at removing the contribution of the extended radio galaxies embedded in the diffuse emission. The final images are produced with the multifrequency deconvolution scheme in WSClean[43,44]. Different resolutions are obtained using different weighting schemes. We did not attempt an in-band spectral analysis because of the narrow frequency range of LoTTS, combined with the uncertainties in the flux density scale and the high r.m.s. noise of the in-band images, which especially affects the resolved low signal-to-noise sources, such as megahalos[8].

## LOFAR low-band antenna data reduction

ZwCl 0634.1+4750 and Abell 665 have been observed with the LOFAR low-band antennas (LBA) for 8 hours in the frequency range of 30–77 MHz and 30–60 MHz, respectively. We used the LBA_OUTER antenna configuration, for which only the outermost antennas are selected, because this simplifies the calibration by reducing the primary beam size and the electromagnetic crosstalk between the dipoles. Observations were performed in multibeam mode, with one beam continuously pointing at the calibrator and one beam continuously pointing at the target.

The data reduction of the calibrator follows the procedure described in ref. [37] and it is used to isolate direction-independent systematic effects such as the bandpass, the stations' clocks drifts and the polarization misalignment caused by time delays between the $X$ and $Y$ polarization signals. The solutions are then applied to the target field along with the primary beam correction in the direction of the target.

The self-calibration steps of the target field are described in ref. [45]. The self-calibration starts with a model obtained from the combination of existing surveys at higher frequencies: TGSS[46], NVSS[47], WENSS[48] and VLSS[49]. We estimate the spectral indices of the sources present in these surveys and we extrapolate their flux densities to LBA frequencies. Then, we estimate direction-independent (field-averaged) differential total electron content solutions by calibrating against the predicted model. Next we solve and correct for the average differential Faraday rotation and second-order beam effects. Sources outside the main lobe of the primary beam are imaged and subtracted from the $uv$-data before proceeding with a second self-calibration cycle. The main errors that still affect the data at this point are the direction-dependent errors caused by the ionosphere. The procedure to correct for direction-dependent errors is discussed in refs. [45,50]. As a first step, sources in the direction-independent calibrated image are grouped by proximity and the brightest groups are identified to be used

as calibrators in the direction-dependent calibration. All the sources in the field are subtracted using the direction-independent model image. We then iterate on the calibrators, starting with the brightest one. The calibrator's visibilities are added back to the data and the dataset is phase-shifted in the calibrator direction. We run several rounds of self-calibration and the improved model of the calibrator is re-subtracted to produce a cleaner empty dataset, before repeating the procedure for the next brightest calibrator.

After the wide-field direction-dependent calibration, further improvement of the image quality can be achieved by extraction and self-calibration of a small region around the target of interest. We follow the idea presented in ref. [41] and use an LBA-specific implementation of the extraction strategy. We use the final model and the solutions of the direction-dependent calibration to accurately subtract all sources in the field except for those in a circular region around the targets. The radii of these regions are 23′ and 14′ for ZwCl 0634.1+4750 and Abell 665, respectively, and they are chosen to include sufficient compact source flux densities to obtain robust calibration solutions. Then the data is phased-shifted to the centre of this region and further averaged in time and frequency. After correcting for the primary beam in this direction, we perform several rounds of self-calibration on the target, starting from the model obtained using the solutions of the closest direction-dependent calibrator. The images at high and low resolution are shown in Extended Data Fig. 1.

## High-resolution 144 MHz images of the clusters and source subtraction procedure

Diffuse emission on scales larger than the radio halos and not associated with radio galaxies is already clearly visible for the four clusters at intermediate resolution (20–30″). To highlight this emission, we removed the contribution of the discrete sources embedded in the diffuse emission by subtracting them out from the $uv$-data. We proceeded in two steps: as a first step, we produced an image at high resolution (6–10″, shown in Extended Data Fig. 2). The model image that we obtained contains the compact sources in the field and part of the extended sources, such as tails of radio galaxies and the central radio halo. However, it does not contain the diffuse emission from regions beyond the radio halo. To produce a source-subtracted image in which both the radio halo and the new emission are clearly visible, we removed the clean components in the region of the radio halo from the model image before we subtracted the rest from the visibilities. These images, for which only the high-resolution source subtraction has been performed, are denoted in Extended Data Table 1 by 'HR'. Then, we imaged these new visibilities at lower resolution (30″ and 60″). The 60″ resolution source-subtracted images are shown in Extended Data Fig. 3b,d,f,h. We note that the large-scale emission does not contain clear residuals or artefacts from the subtracted sources. We inspected the model images obtained at 30″ resolution and we retained only the clean components associated with the radio halo and what was left of the extended radio galaxies. Then we subtracted them from the $uv$-data. Finally, we produced an image at very low resolution (2′, Fig. 1), which only contains the residual large-scale emission. The images in which the radio halo has also been subtracted are denoted in Extended Data Table 1 by 'HR+RH'.

## Surface brightness radial profiles

Following the approach described in ref. [9,13], we derived the azimuthally averaged surface brightness radial profile of the radio emission of the four clusters (Extended Data Fig. 3). We used low-resolution (60″) images after subtracting only the sources visible in the high-resolution image to have a good compromise between sensitivity to the extended emission and resolution to characterize the profile, and to possibly distinguish between the radio halo and the more extended emission. We note that the whole extension of the diffuse emission is best detected in the 2′ resolution images (Fig. 1) that we used to measure

the size of the sources. If the image contains some residuals from bright diffuse sources, we masked them and we excluded the masked pixels when calculating the surface brightness. This is the case in Abell 665, for which we masked the residuals of a diffuse patch of emission that does not appear to be associated with the megahalo. In Abell 697 we also masked the residuals of a bright compact source in the north and a radio galaxy in the south (these sources are visible in Extended Data Fig. 2). For Abell 665 we considered only the southern part of the cluster because the northern part has been crossed by a shock front[51], which may have altered the properties of the diffuse emission. We averaged the radio brightness in concentric annuli, centred on the peak of the radio halo and chose the width of the annuli to be half of the full-width half-maximum of the beam of the image. We considered only annuli with an average surface brightness profile higher than three times the uncertainty associated with the annuli surface brightness. In the images we show the detection limits for each annulus calculated as r.m.s. $\times \sqrt{N_{\text{beam}}}$, where $N_{\text{beam}}$ is the number of beams in the annulus.

All the profiles show a discontinuity. The central annuli before the discontinuity follow an exponential profile, similar to other classical radio halos[9,13]. This first component can be fitted with an exponential law in the form:

$$I(r) = I_0 e^{-\frac{r}{r_e}}, \tag{1}$$

where $I_0$ is the central surface brightness and $r_e$ is the $e$-folding radius, that is the radius at which the surface brightness is $I_0/e$.

To perform the fit, we first generated a model using equation (1) with the same size and pixel size of the radio image and we convolved it with a Gaussian with a full-width half-maximum equal to the beam of the image. Then, we azimuthally averaged the exponential model with the same set of annuli used for the radio halo. The resulting surface brightness profile is the fit that takes into account the resolution of the image and the uncertainties associated with the sampling of the radial profile.

The discontinuity in surface brightness is less pronounced in Abell 665 than in the other three cases. However, there is a second component that is not consistent with the radio halo profile. In addition, the large-scale diffuse emission in Abell 665 shows clear differences with the central radio halo also in terms of spectrum and emissivity (see below). Hence, we consider it a megahalo.

In two cases, ZwCl 0634.1+4750 and A697, the diffuse emission beyond the classical radio halos is not symmetric with respect to the centre of the radio halo. Therefore, we performed the fit also in the semi-annuli on the left-hand side of the red line shown in Extended Data Fig. 3. The discontinuity is also present in this case.

## Spectral index analysis

We measured the spectral index of the radio halo in ZwCl 0634.1+4750 in the central region shown in Extended Data Fig. 1a. We obtained a flux density of 39.6 ± 4.0 mJy at 144 MHz and 138.3 ± 15.9 mJy at 53 MHz, corresponding to $\alpha = -1.25 \pm 0.15$. The uncertainties on the integrated flux densities in this section take into account the systematic error given by the uncertainty on the flux scale and the statistical error associated with the r.m.s. noise of the image in the integration area. The emission beyond the radio halo in ZwCl 0634.1+4750 is marginally detected by the LBA. Therefore, we focused on the region shown in Extended Data Fig. 1a to estimate the integrated spectral index of this new emission. In this region we measured 35.8 ± 7.5 mJy at 53 MHz and 6.8 ± 0.6 mJy at 144 MHz, which gives a spectral index $\alpha = -1.62 \pm 0.25$. The average surface brightness of the megahalo at 144 MHz, extrapolated at 53 MHz with this spectral index, would be below two times the r.m.s. noise of the lowest resolution LBA image. This explains why we do not detect the whole megahalo emission in the LBA.

We used the HBA and LBA images at 37″ resolution to produce the spectral index map of Abell 665 (Extended Data Fig. 4). In these images,

the sources visible in the high-resolution image have been subtracted. We produced a pixel-by-pixel spectral index map using all pixels that had a surface brightness above $2\sigma$ in both images. We then carried out the linear regression using a bootstrap Monte-Carlo method obtaining 1,000 estimations of the spectral index values per pixel. The reported spectral index is the mean of the distribution of the estimations and the uncertainty is its standard deviation.

The spectral index of the diffuse emission in Abell 665 ranges from around −0.5 to −2. In particular, we note that the spectrum of the northern part is relatively flat. However, a combination of shock and turbulent acceleration could produce a flatter and less uniform spectrum than expected from turbulence alone[51]. Hence, we focused on the southern part of the cluster, marked by the regions in Extended Data Fig. 1 to derive the integrated spectral index. The choice of the regions is based on the surface brightness radial profile (Extended Data Fig. 3c,d). In particular, the limit between the regions where we measure the flux densities corresponds to the annulus in which the surface brightness profile flattens with respect to the classical radio halo exponential function. In these regions we measured a flux density for the radio halo of 120.3 ± 12.1 mJy at 144 MHz and 614.6 ± 62.9 mJy at 44 MHz, corresponding to $\alpha = -1.39 \pm 0.12$. In the region beyond the radio halo, we measured 58.2 ± 6.1 mJy at 144 MHz and 398. ± 45.8 mJy at 44 MHz, obtaining a spectral index $\alpha = -1.64 \pm 0.13$.

## Emissivity

We calculated the volume-averaged emissivity at frequency $v$ in radio halos and megahalos by assuming that their radio power, $P_v$, comes from a sphere of radius $R$:

$$J_v = \frac{P_v}{\frac{4}{3}\pi R^3} \tag{2}$$

We estimated the source radii via $\sqrt{R_{\min} \times R_{\max}}$ (ref. [52]), where $R_{\min}$ and $R_{\max}$ are the minimum and maximum radii of the $3\sigma$ contours, respectively, in Fig. 1 (we used the 30″ resolution images for radio halos and the 2′ resolution images for megahalos). We subtracted the extended radio galaxies in the fields to the best of our abilities given current techniques. However, we are aware that the central regions of the 2′ resolution images shown in Fig. 1 may be affected by residuals from the subtracted radio halos. It seems reasonable to assume that the megahalo also permeates the region of the classical radio halos. Hence, to estimate the flux density of the new type of emission, we measured the mean surface brightness in a region that excludes the central halos and then multiplied it by the total area within the $3\sigma$ contours. A direct measurement of the flux density inside the area delimited by the $3\sigma$ contours would give marginal differences of the order of 5–15%. To compare these emissivities with the typical emissivity of radio halos[24], we estimated the flux density at 1.4 GHz, by assuming a conservative spectral index of −1.3, for both radio halos and megahalos. If the spectral index of the detected emission is steeper, the emissivity at 1.4 GHz would be even lower. The emissivities of radio halos and megahalos for each cluster are listed in Extended Data Table 2, together with the size and the spectral index (when available) of the sources. The uncertainties include the systematic errors given by the uncertainty on the flux scale, the statistical error associated with the r.m.s. noise of the image in the integration area and the subtraction error related to the uncertainty on the subtracted flux. For the latter we used the approach described in ref. [42]. We do not take into account the uncertainty in the estimated size of the diffuse emission. The emissivity of megahalos is a factor of around 20–25 lower than the emissivity of the radio halos in the same clusters. For comparison, the typical emissivity of classical radio halos at 1.4 GHz ranges between $5 \times 10^{-43}$ and $4 \times 10^{-42}$ erg s$^{-1}$ cm$^{-3}$ Hz$^{-1}$ in clusters with similar masses[24].

## Simulations of turbulence in galaxy clusters

High-resolution hydrodynamic, cosmological simulations of the ICM can shed some light on megahalos. We analysed a set of 20 ($M_{500} \geq 3 \times 10^{14} M_\odot$ at $z = 0.0$) galaxy clusters[22] and used small-scale filtering to calculate the turbulent kinetic energy flux, $F_{turb} = \rho\, \sigma_v^3/L$, within simulated cells (each cell having a $32^3$ kpc$^3$ volume), where $\rho$ is the local gas density and $\sigma_v$ is the dispersion of the velocity field measured within the scale length $L$ (ref. [53]). Then we measured the average distribution of $F_{turb}$ in relaxed and disturbed clusters, and in the inner as well as outer regions, where megahalos are observed (approximately $0.4 \times R_{500} < r \leq R_{500}$).

Extended Data Fig. 5 shows that the turbulent kinetic energy flux, $F_{turb}$, in the central regions of postmerger clusters is elevated by a factor of around 10 compared with relaxed clusters. This is consistent with the idea that radio halos are produced by the dissipation of turbulence following a merger[18,19,54–56]. However, in the outer regions $F_{turb}$ is notably similar in relaxed and disturbed clusters. This shows the presence of a baseline level of turbulence that is induced by the continuous accretion of matter in cluster outskirts. Hence, this level of turbulence is likely to be common to all clusters. In this picture, megahalos may also be generated by Fermi II re-acceleration in more relaxed clusters without central radio halos. Future deeper observations, such as those that will be made with LOFAR 2.0, will enable us to test this scenario.

## Data availability

The radio observations are available in the LOFAR Long Term Archive (https://lta.lofar.eu/).

## Code availability

The codes used for the LOFAR HBA data reduction are available at https://github.com/lofar-astron/prefactor and https://github.com/mhardcastle/ddf-pipeline. The codes used for the LOFAR LBA data reduction are available at https://github.com/revoltek/LiLF. The cosmological simulations were produced using the ENZO cosmological grid code, available at enzo-project.org.

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

**Acknowledgements** V.C. acknowledges support from the Alexander von Humboldt Foundation. F.d.G. and M.B. acknowledge support from the Deutsche Forschungsgemeinschaft under Germany's Excellence Strategy – EXC 2121 'Quantum Universe' – 390833306. F.V. acknowledges financial support from the H2020 initiative through the ERC StG MAGCOW (grant no. 714196). F.V. gratefully acknowledges the Gauss Centre for Supercomputing e.V. (www.gauss-centre.eu) for funding this project by providing computing time through the John von Neumann Institute for Computing on the GCS Supercomputer JUWELS at Jülich Supercomputing Centre (JSC), under project RADGALICM/2 (grant nos 22552 and 25063). R.C., G.B. and F.G. acknowledge support from Istituto Nazionale di Astrofisica (INAF) mainstream project 'Galaxy Clusters Science with LOFAR' 1.05.01.86.05. G.D.G. acknowledges support from the Alexander von Humboldt Foundation. A.D. acknowledges support by the BMBF Verbundforschung under the grant no. 05A20STA. R.J.v.W. acknowledges support from the ERC Starting Grant ClusterWeb 804208. The Jülich LOFAR Long Term Archive and the German LOFAR network are both coordinated and operated by the JSC, and computing resources on the supercomputer JUWELS at JSC were provided by the Gauss Centre for supercomputing e.V. (grant no. CHTB00) through the John von Neumann Institute for Computing. LOFAR[5] is the LOw Frequency ARray designed and constructed by ASTRON. It has observing, data processing and data storage facilities in several countries, which are owned by various parties (each with their own funding sources), and are collectively operated by the ILT foundation under a joint scientific policy. The ILT resources have benefitted from the following recent major funding sources: CNRS-INSU, Observatoire de Paris and Université d'Orléans, France; BMBF, MIWF-NRW, MPG, Germany; Science Foundation Ireland, Department of Business, Enterprise and Innovation, Ireland; NWO, the Netherlands; The Science and Technology Facilities Council, United Kingdom; Ministry of Science and Higher Education, Poland; and INAF, Italy. This research made use of the Dutch national e-infrastructure with support of the SURF Cooperative (e-infra 180169) and the LOFAR e-infra group; the LOFAR-IT computing infrastructure supported and operated by INAF; and the Physics Department of Turin University (under the agreement with Consorzio Interuniversitario per la Fisica Spaziale) at the C3S Supercomputing Centre, Italy. This research made use of the University of Hertfordshire high-performance computing facility and the LOFAR-UK computing facility located at the University of Hertfordshire and supported by STFC (grant no. ST/P000096/1); the Italian LOFAR-IT computing infrastructure supported and operated by INAF; and the Physics Department of Turin university (under an agreement with Consorzio Interuniversitario per la Fisica Spaziale) at the C3S Supercomputing Centre, Italy.

**Author contributions** V.C. coordinated the research project, produced and analysed the radio images and wrote the manuscript. F.d.G. performed the LOFAR LBA data reduction and analysis and helped with the writing of the manuscript. M.B., F.V. and G.B. helped with the writing of the manuscript and with the theoretical interpretation of the results. F.V. performed the numerical simulation. H.W.E. contributed to the analysis of the LOFAR LBA data and gave feedback on the manuscript. T.W.S. leads the LoTSS survey, coordinated the LOFAR data reduction and helped with the writing of the manuscript. R.J.v.W. helped with the writing of the manuscript. A.B. contributed to the LOFAR HBA data reduction and gave feedback on the manuscript. R.C., G.D.G., F.G. and H.J.A.R. gave feedback on the manuscript. A.D. contributed to the LOFAR HBA data reduction. C.T. contributed to the development of the data reduction software. All the authors of this manuscript are members of the LOFAR Surveys Key Science Project.

**Funding** Open access funding provided by Universität Hamburg.

**Additional information**
**Correspondence and requests for materials** should be addressed to V. Cuciti.

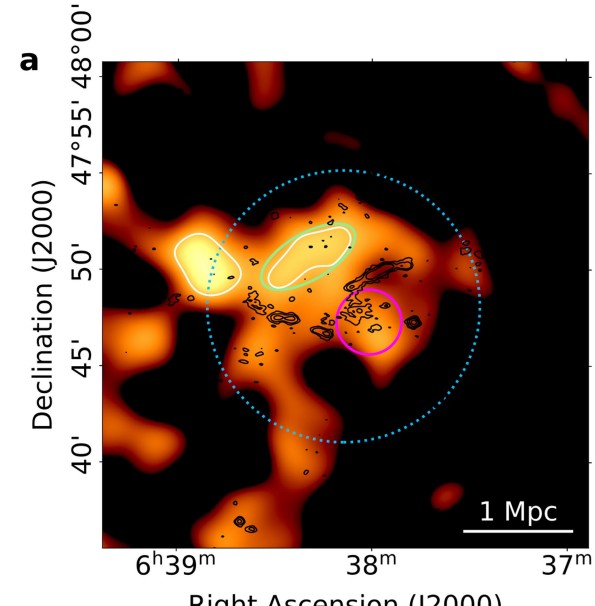

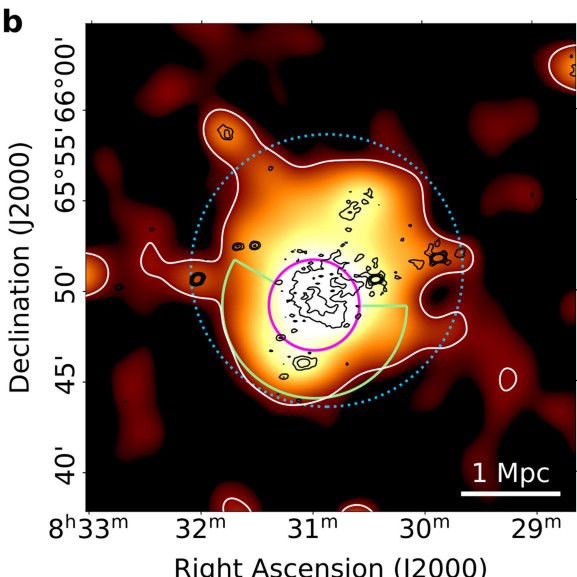

**Extended Data Fig. 1 | LOFAR LBA images.** a) ZwCl 0634.1+4750, b) Abell 665. Orange images and white contours: LOFAR data at 2′ resolution after subtraction of all sources, including the radio halo. The contour is at 3$\sigma$. Black contours: high-resolution data (18.1″ × 18.1″ for ZwCl 0634.1+4750 and 23″ × 14″ for Abell 665), contours at (3, 6, 12, 24) × $\sigma$. The magenta and green regions mark the area used to calculate the integrated spectral index of the radio halos and of the newly discovered emission, respectively. The radius of the light blue circle corresponds to $R_{500}$.

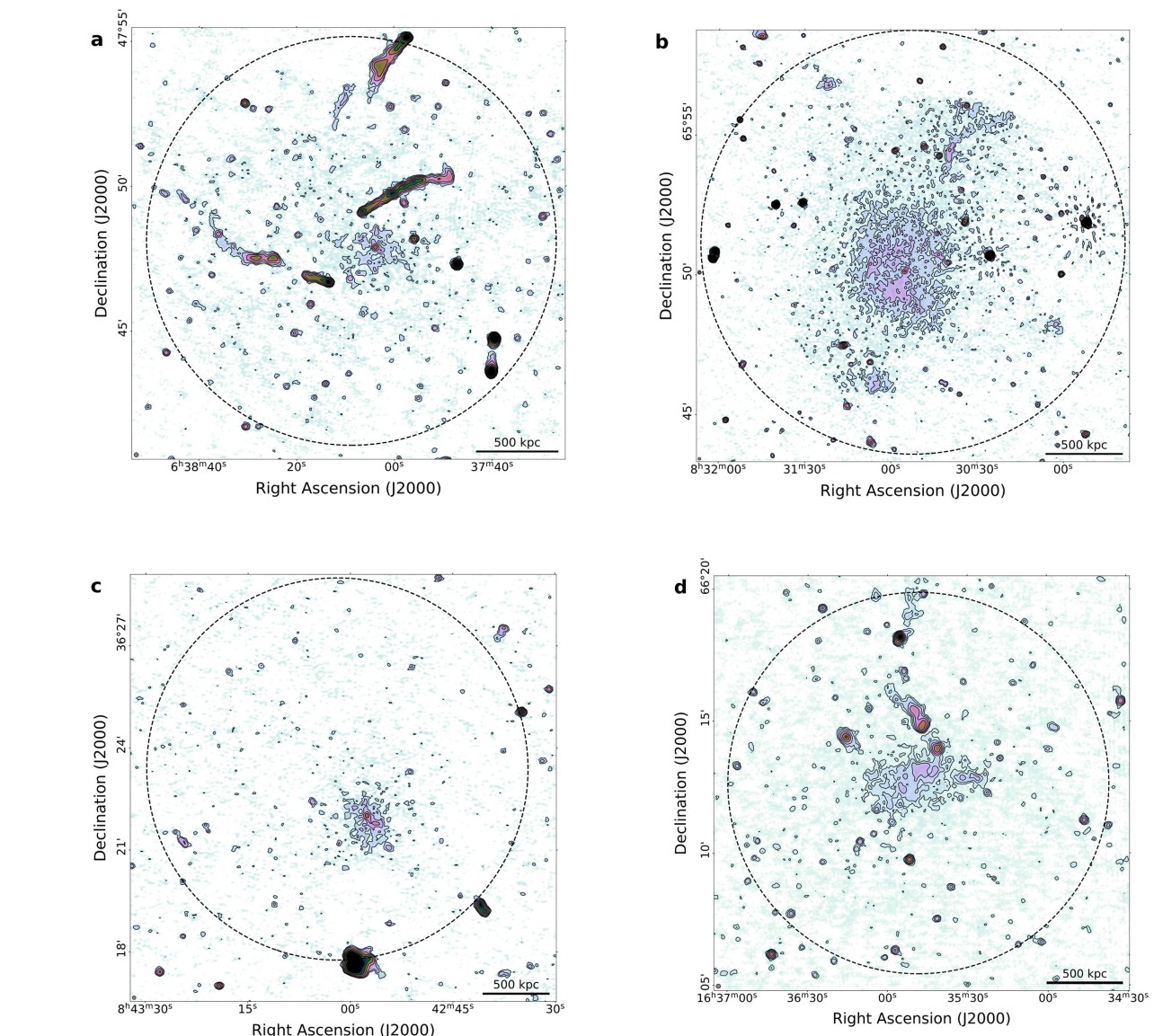

**Extended Data Fig. 2 | High-resolution LOFAR 144 MHz images.** ZwCl 0634.1+4750 (a, 9.1″ × 9.1″), Abell 665 (b, 6.3″ × 6.3″), Abell 697 (c, 6.6″ × 6.6″), Abell 2218 (d, 10.1″ × 10.1″). Contours start at 3$\sigma$ and they are spaced by a factor 2. The radius of the dashed circle corresponds to $R_{500}$.

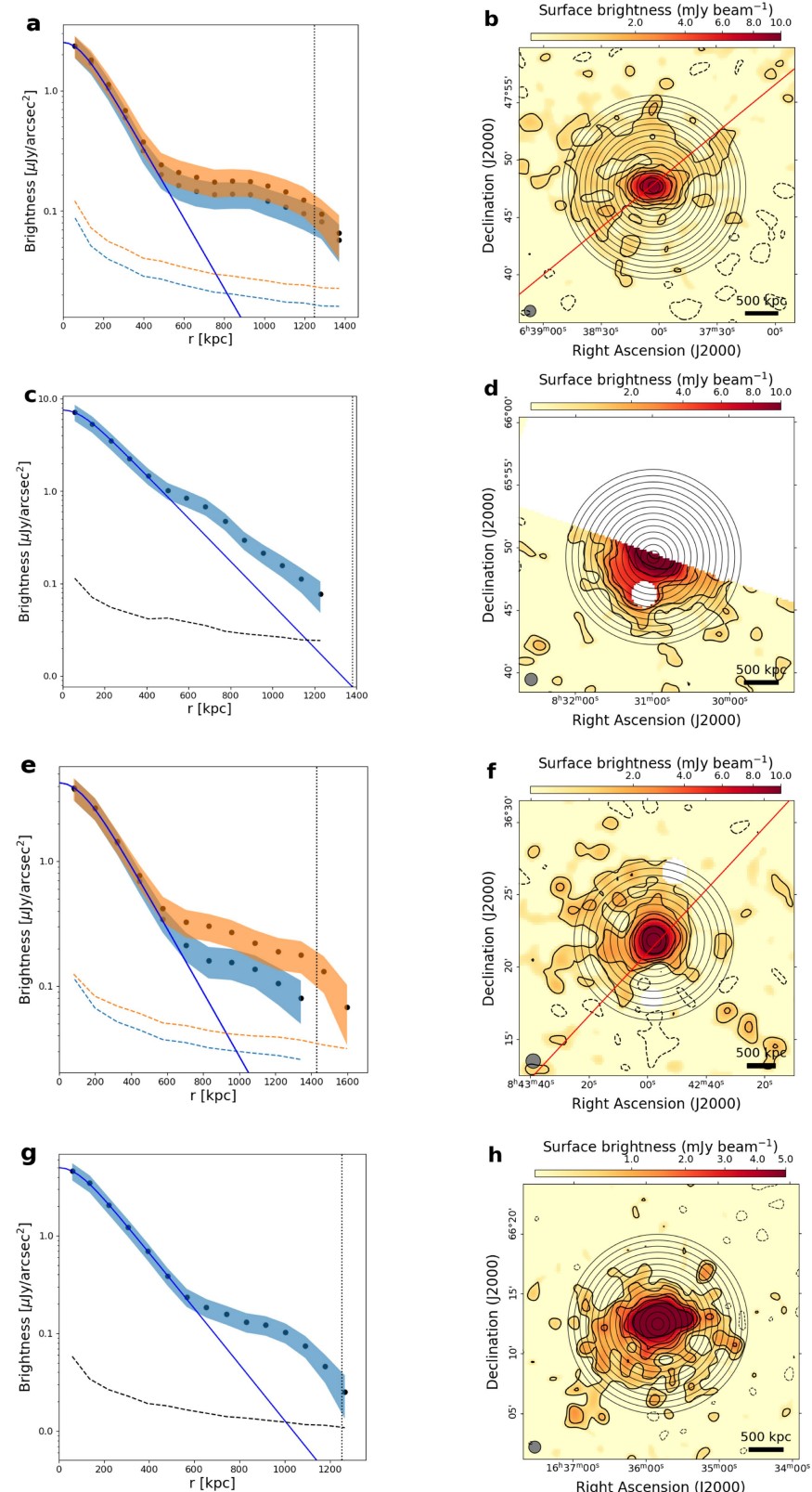

**Extended Data Fig. 3 | Surface brightness radial profiles.** From top to bottom: ZwCl 0634.1+4750, Abell 665, Abell 697, Abell 2218. a, c, e, g: surface brightness radial profile measured on the regions shown on the maps on the right. The blue line is the exponential fit to the radio halo data. The dashed lines mark the 1σ detection limit for each annulus (or semi-annulus). The vertical line marks $R_{500}$. Shaded regions represent the 1σ uncertainty. For ZwCl 0634.1+4750 and Abell 697 we derived the profiles in the entire annuli (blue shaded region) and only in the semi-annuli on the left hand side of the red line shown on the right panels (orange shaded region). b, d, f h: 144 MHz images at 60″ resolution after subtraction of the sources visible in the high-resolution images.

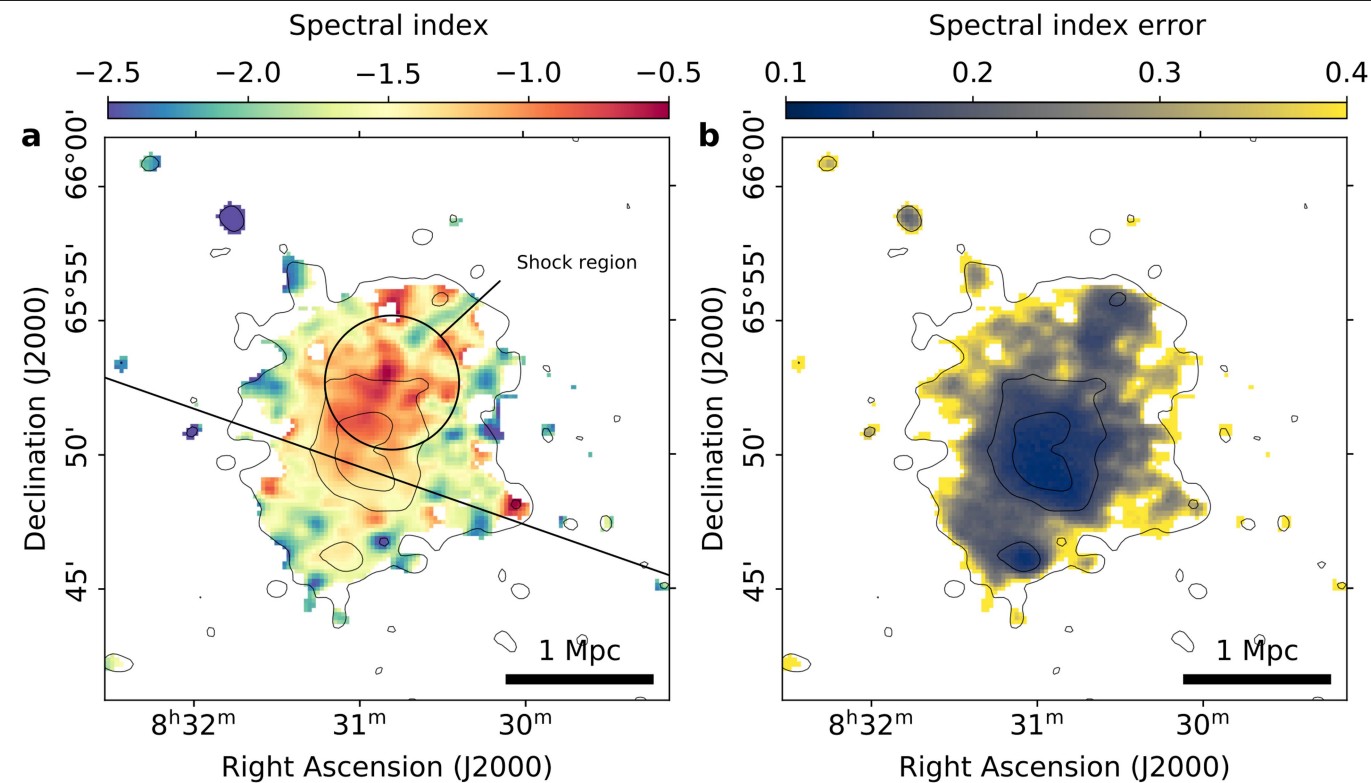

**Extended Data Fig. 4 | 44-144 MHz spectral index map of Abell 665.** a) Spectral index map. The black lines delimits the "southern part" of the cluster used for the surface brightness profile in Extended Data Fig. 2. b) 1$\sigma$ error associated with the spectral index map.

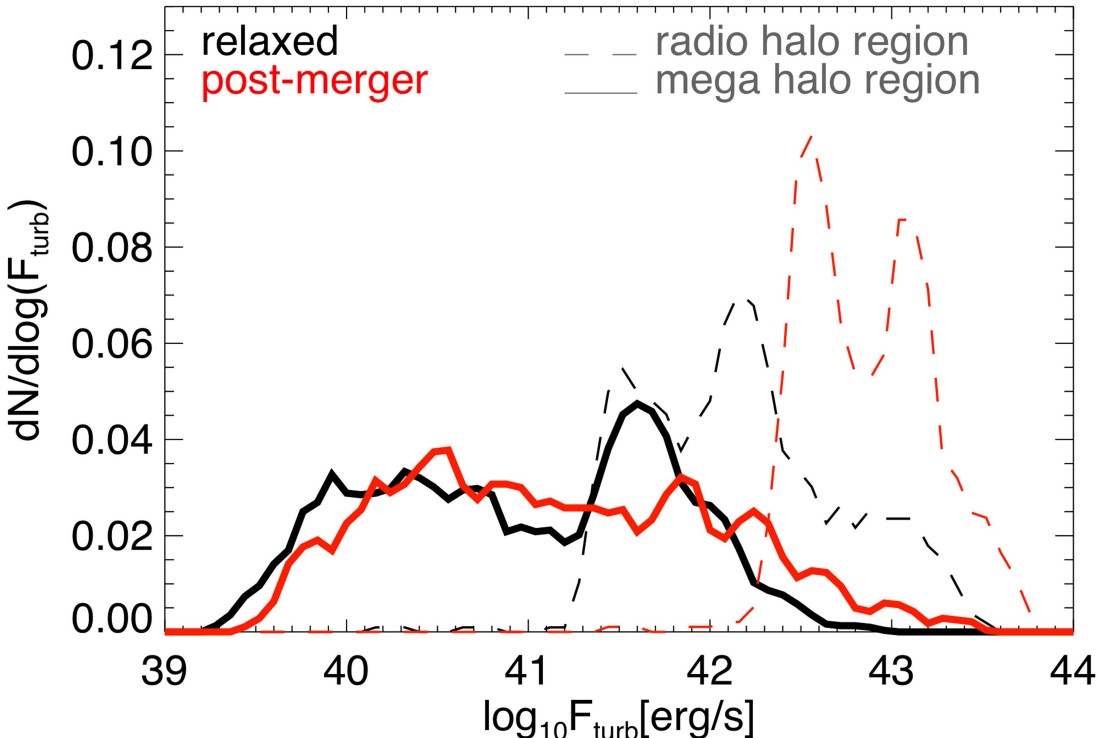

**Extended Data Fig. 5 | Simulated turbulence.** Probability distribution function of the turbulent kinetic energy flux in post-merger or relaxed clusters simulated by Vazza et al. (2011)[22], either within the region of classic radio halos (dashed line), or within the outer region (solid line) where the newly discovered megahalos are located.

**Extended Data Table 1 | Properties of the LOFAR images**

| Cluster | Frequency (MHz) | Weighting Briggs | Taper | Resolution (″×″) | rms noise (mJy/b) | Subtraction | Fig |
|---|---|---|---|---|---|---|---|
| ZwCl 0634.1+4750 | 144 | −0.1 | – | 9.1 × 9.1 | 0.085 | HR | EDF 2 |
| | | −0.5 | 30″ | 30 × 30 | 0.18 | HR | Fig. 1 Fig. 2 |
| | | −0.2 | 60″ | 60 × 60 | 0.3 | HR | EDF 3 |
| | | −0.5 | 2′ | 120 × 120 | 0.42 | HR+RH | Fig. 1 Fig. 2 |
| | 53 | −0.3 | – | 18.1 × 18.1 | 1.3 | – | EDF 1 |
| | | 0.0 | 2′ | 120 × 120 | 5.0 | HR+RH | EDF 1 |
| Abell 665 | 144 | −0.5 | – | 6.3 × 6.3 | 0.06 | HR | EDF 2 |
| | | −0.5 | 30″ | 30 × 30 | 0.17 | HR | Fig. 1 |
| | | −0.2 | 60″ | 60 × 60 | 0.3 | HR | EDF 3 |
| | | −0.5 | 2′ | 120 × 120 | 0.5 | HR+RH | Fig. 1 |
| | 44 | −0.3 | – | 23 × 14 | 2.2 | – | EDF 1 |
| | | −0.3 | 2′ | 140 × 121 | 8.0 | HR+RH | EDF 1 |
| | | −0.3 | 25″ | 37 × 37 | 2.2 | HR | EDF 4 |
| Abell 697 | 144 | −0.5 | – | 6.6 × 6.6 | 0.09 | – | EDF 2 |
| | | −0.5 | 30″ | 30 × 30 | 0.26 | HR | Fig. 1 |
| | | −0.5 | 60″ | 60 × 60 | 0.39 | HR | EDF 3 |
| | | −0.5 | 2′ | 120 × 120 | 0.5 | HR+RH | Fig. 1 |
| Abell 2218 | 144 | −0.1 | – | 10.1 × 10.1 | 0.07 | HR | EDF 2 |
| | | −0.5 | 30″ | 30 × 30 | 0.13 | HR | Fig. 1 |
| | | −0.5 | 60″ | 60 × 60 | 0.2 | HR | EDF 3 |
| | | −0.5 | 2′ | 120 × 120 | 0.32 | HR+RH | Fig. 1 |

**Extended Data Table 2 | Properties of radio halos and megahalos**

| Cluster name | Source | Radius (kpc) | $J_{1.4GHz}$ (erg s$^{-1}$ cm$^{-3}$ Hz$^{-1}$) | $\alpha_{LBA}^{HBA}$ |
|---|---|---|---|---|
| ZwCl 0634.1+4750 | RH | $352 \pm 32$ | $(3.2 \pm 0.2) \times 10^{-43}$ | $-1.25 \pm 0.15$ |
| | MegaH | $1400 \pm 130$ | $(1.3 \pm 0.4) \times 10^{-44}$ | $-1.62 \pm 0.25$ |
| Abell 665 | RH | $454 \pm 13$ | $(1.2 \pm 0.2) \times 10^{-42}$ | $-1.39 \pm 0.12$ |
| | MegaH | $1417 \pm 70$ | $(6.3 \pm 0.8) \times 10^{-44}$ | $-1.64 \pm 0.16$ |
| Abell 697 | RH | $480 \pm 43$ | $(6.4 \pm 3.0) \times 10^{-43}$ | $-$ |
| | MegaH | $1432 \pm 50$ | $(2.4 \pm 0.6) \times 10^{-44}$ | $-$ |
| Abell 2218 | RH | $585 \pm 83$ | $(2.4 \pm 0.5) \times 10^{-43}$ | $-$ |
| | MegaH | $1333 \pm 102$ | $(1.2 \pm 0.3) \times 10^{-44}$ | $-$ |