## [Peer Review File · Nature]

Manuscript Title: Galaxy clusters enveloped by vast volumes of relativistic electrons

Reviewer Comments & Author Rebuttals

Reviewer Reports on the Initial Version:

Referees' comments:

Referee #1 (Remarks to the Author):

This paper reports the discovery of a potential new class of diffuse cluster radio emission, dubbed 'megahalos' by the authors, using LOFAR HBA observations of four massive, nearby Planck clusters known to host typical radio halos. The megahalo's low surface brightness emission spans much larger regions of the cluster, extending out to R500, and has different surface brightness profile characteristics to the hosted radio halo. For the two systems with LOFAR LBA detections, the megahalo is found to be steeper spectrum than the hosted radio halo, and all four megahalos are determined to have substantially lower emissivity than typical radio halos in similar mass systems. These findings lead to the conclusion that a different physical mechanism may be contributing to the formation of these megahalos, as compared to radio halos, and therefore may represent a new class of diffuse cluster radio emission.

The paper is fairly well-written, though the main section suffers from repetition in places. The methodologies used are sound and the conclusions are convincing, well-supported by the results. The findings are of significant interest to the cluster community and are sufficiently original to warrant publication in Nature. I therefore recommend acceptance of the paper after minor revision. My main comments on the manuscript, detailed below, are mostly to improve clarity for the reader.

>> While reading the main section, I wondered whether the megahalos shown in Fig 1 could be contaminated by the hosted radio halo, i.e., RH emission being smoothed out to some extent. This is answered much later in the methodologies section, however I suggest making it clear when introducing and discussing Fig 1 that these images are after subtraction of the radio halo emission, in addition to the more typical source subtraction (at least in the figure caption). In general, as there are several types of source subtraction performed in the analysis, being more specific as to the relevant subtractions performed when mentioned "source subtracted" will assist the reader in following the discussion and results in the main paper.

>> The spectral index of the megahalos, although steeper than the hosted radio halo, are in line with those found for ultra-steep spectrum radio halos. There is no discussion as to whether these classes may be related, or be related to similar physical mechanisms.

>> Related to the possibility of this being a new class, it would be useful to report the expected megahalo 1.4 GHz flux density range for the four clusters, and discuss this in relation to the power-mass correlation for the given mass range

>> In several places 'error' is used when 'uncertainty' is more appropriate – if error values are known, they can be corrected for.

>> References: paper shows a strong preference for referencing co-author work. Although I acknowledge that several coauthors are leaders in the diffuse emission fields, more effort could be made to cite more broadly and acknowledge the work of other teams/individuals.

>> Figure 2: suggest changing the colour of the 30arcsec resolution halo – currently it's the same as the lowest brightness of the megahalos, which may be aesthetically pleasing but can be confusing for image interpretation. A grey region may be appropriate instead.

>> Figure 2: Nowhere is it discussed what the inset image is. After reading the full document, it's clear that this is the same image as shown in EDF3, however the image details should be given in the Fig 2 caption.

>> line 108: "of maintaining"  "for maintaining"

>> line 122: "with what proposed"  "with what is proposed"

>> lines 128-129: why would the macro/micro-physics change abruptly?

>> Table 1: why are there different robust weighting iterations across clusters? How are the optimal weightings chosen?

>> EDF1: Why are different sigma limits used for the two clusters (2sig for ZwCl0634 and 3sig for A665)? Sigma limits for interpretation should be uniformly chosen so as not to create bias.

>> EDF1: The red and green regions are difficult to see in the left panel, particularly the red one. Suggest changing the red region to magenta or cyan in both cases.

>> line 320: Why are different angular scales chosen (45' vs 28')? Are these related to a physical scale at the cluster redshift, or is it arbitrary/linked to observed emission size?

>> lines 326-327: "clearly visible at 20as – 30as" – I disagree, as this is not apparent in Fig 1 (black contours) except perhaps for ZwCl0634. If this is the wrong figure to be comparing to, tell the reader which figures show the "clearly visible" emission (EDF2 perhaps?).

>> lines 348-349: "detected at best" – reword for better clarity, currently reads like the emission is not really visible but best seen at 2 arcmin resolution.

>> EDF2: Suggest adding the R500 region to each panel so that the high resolution imaging can more readily be visually compared to the other low res images presented

>> EDT1: might be good to add a column with details about the relevant source subtraction, e.g.,

none, HR only, HR + RH. Related to an earlier comment, the phrase “source-subtracted” in the paper is ambiguous given the number of different subtractions used. Also might be useful to add a note for those images presented in other figures, to help the reader correlate the results between different parts of the paper, i.e., “the 30”, 60” and 2’ HBA images are presented in Figures XX, YY, and ZZ, respectively.”

>> EDF3: right panels: note in the caption that these are after subtraction of the HR components only and include the RH emission.

>> EDF3: A665 left – the discontinuity is not as clear as for the other profiles and it is worth adding some discussion in the text as to why this may be

>> What is the bandwidth of the LOFAR 144 MHz data? Is it possible to obtain in-band spectral index maps for the megahalos? (if no, despite bandwidth availability, mention/discuss)

>> EDF4: add boundary line to denote “southern part”

>> EDF4: is there any spatial correlation between where the radio halo lies and any changes in the spectral index map?

>> lines 395-396: “bottom panels” – those are for A2218, the A665 panels are in the second row

>> EDT2: keep the same order as in all the other figures and tables in the paper (ZwCl, A665, A697, A2218)

>> line 403: “best of our possibilities”  “best of our abilities given current techniques”

>> Emissivity section: include the equation used to calculate the emissivity, rather than just including a paper reference.

Referee #2 (Remarks to the Author):

A. The paper presents the discovery of a new kind of radio source associated with clusters of galaxies. It has only now been possible to make this discovery thanks to the sensitivity to large-scale diffuse radio emission of LOFAR. More specifically, the authors show evidence of a large “megahalo” structure surrounding 4 individual clusters of galaxies that appears to have distinct radio properties compared to previous known radio structures. These detections are robust and novel.

B. The paper is well-written, well-justified and the results are clear and extremely intriguing. These discoveries are not only important for the galaxy cluster community, but also have broader implications for several other subfields in astrophysics (for example, the astroparticle community,

large-scale structure community, high-energy community, etc). It is novel and I am very happy to support its publication in a journal such as Nature.

However, I do have some technical/scientific issues that I think need to be addressed before the paper can be accepted.

C. Data & Methodology (technical/scientific issues):

1) Sample selection: Fig. 3 is an important plot that shows a key result, namely that LOFAR is identifying just the tip of the iceberg of these “megahalo” sources. However, for this statement to hold, Fig. 3 needs to show a well-defined and well-constructed sample of clusters for which the authors have uniform data. This is in fact the case of the Planck clusters in the LOFAR data release 2, for which the authors understand very well the data limitations and quality. However, by adding ZwCl 0634.1 to the sample, it breaks down the argument. I understand that the megahalo discovered in this object highlights a “proto-typical” example, but I believe that the paper could be just as strong, and even stronger if the authors limit themselves to the Planck sample. It would make the results statistically complete.

Along these lines, if the authors feel strongly on keeping ZwCl 0634.1 in their paper, and that they are aware that Abell 2255 has a similar radio megahalo (as stated in line 159), then they should include Abell 2255 in their paper. Also in what context was ZwCl 0634.1 observed with LOFAR and how was its megahalo discovered? This needs to be explained/clarified in more detail.

2) In the paper, the authors make a strong claim that radio megahalos must be a distinct phenomenon from radio halos. This is a very important claim, which I do not believe the paper justifies. The authors demonstrate well that the properties are different (i.e. lower emissivity, different emissivity profile, steeper spectrum), but this does not imply that they must be of different origin or an entirely new phenomenon. In other words, their observations do not (yet) demonstrate that the seed population or the acceleration process must be completely different.

If the authors want to make this claim, they must demonstrate it.

For example, can they produce numerical simulations that support this claim? Or do they have theoretical arguments that predict the existence of a distinct class of extended radio structures (versus regular halos)? Couldn't radio megahalos simply be an older, more diffuse population of radio halos? In this case, this does not make them distinct necessarily.

Also, lines 106 to 109, how is turbulence operating in a different regime? This needs to be clarified and justified quantitatively. Similarly, in lines 127 to 139, these are important claims, but without simulations or calculations to support them.

Related to this point, according to Extended Data Figure 3, Abell 665 appears to have a profile similar

to its radio halo, so the distinction may not be as clear as claimed.

Overall, if the authors feel strongly that radio megahalos are indeed a distinct class of radio sources, they need to provide additional evidence.

3) Minor comment related to Figure 2: The authors need to add a section in the Supplement series on how the gas density (computer simulation) was obtained and selected to match that of ZwCl 0634 and why that particular orientation was chosen. The authors should also add a warning note in the caption specifying that the figure is just for illustrative purposes to illustrate the extent expected of the gas density versus the megahalo (given that the computer simulation was not specifically tuned for ZwCl 0634).

D & E. Robust results with appropriate uncertainties for the field.

F. Suggested improvements:

1) Figure 2 (as well as lines 98 to 101) and Figure 3: if LOFAR is detecting just the tip of the iceberg, is there a possibility that the megahalos actually extend further beyond than what the authors are able to detect currently with LOFAR? According to Extended Data Figure 3, the profiles appear to converge, so one could fit a profile and determine the total extent of the halo (out to about $r=1.5$ Mpc according to Extended Data Figure 3). How does this compare to the distribution of relativistic particles in computer simulations? And will LOFAR 2.0 or SKA be able to detect this? I would invite the authors to add a few sentences on this.

2) Figure 3: I would suggest adding a line (perhaps $\beta=1$) for LOFAR 2.0 and SKA, to illustrate what will be accessible in the near future.

3) Extended Data Table 2: I would suggest adding a bit more information that would be useful to the community. In particular, a column indicating the emissivity of the radio halos (so that the reader can directly compare to the megahalo) and perhaps other properties such as extent of the halo versus megahalos, spectral index, etc. I leave the authors to decide, but I believe such a table would be beneficial for the reader.

4) Figure 1: I suggest making the figures larger to better illustrate the halos (they appear small on printed paper). Perhaps two columns by two rows would better highlight them. I also suggest adding the names of each cluster in the figures to make it easier for the reader.

5) Since the publication is intended for Nature, it would be worth emphasizing the implications of the results for the broader community (beyond the cluster community). This should be added at the end of the main document.

G. Appropriate references.

H. Clarity :

1) When reading the abstract, I had initially understood that the authors had discovered a large radio halo surrounding 4 clusters, and not large radio halos surrounding 4 individual clusters of galaxies. I would rephrase the sentence “we report the first detection.... halos.”

2) “These faint sources are called...Megaparsec.”. The authors need to rephrase the sentence because radio halos are not the only sources – radio mini-halos and relics are also known to exist.

3) Sometimes you cite Zwcl 0634 and sometimes Zwcl0634 and sometimes ZwCl 0634. I would make everything uniform and verify the rest of the cluster names as well.

4) Line 94 to 96: use to correct spectral indexes determined in the supplement series (with uncertainties) and not the approximate values.

5) Figure 2: In the caption, the authors should also specify that the insert (right panel) is at lower resolution of 60” (compared to Figure 1). I could not understand why it looked different compared to the image overlaid on the computer simulation until reaching the supplement series.

6) Line 50: specify the central regions of clusters in kpc.

7) Line 66: for the term distinct, justify that it is distinct or refer to later on (where the authors demonstrate the different properties).

8) Line 350: how were these diffuse bright sources identified and do these include head-tail sources, etc. If so, please clarify.

Author Rebuttals to Initial Comments:

Response to referee 1

The authors would like to thank the referee for the careful reading of the manuscript and the helpful suggestions that improved the quality and readability of the paper.

We list below the authors' reply to each comment raised by the referee.

>> While reading the main section, I wondered whether the megahalos shown in Fig 1 could be contaminated by the hosted radio halo, i.e., RH emission being smoothed out to some extent. This is answered much later in the methodologies section, however I suggest making it clear when introducing and discussing Fig 1 that these images are after subtraction of the radio halo emission, in addition to the more typical source subtraction (at least in the figure caption). In general, as there are several types of source subtraction performed in the analysis, being more specific as to the relevant subtractions performed when mentioned "source subtracted" will assist the reader in following the discussion and results in the main paper.

A: We specified the source subtraction that we performed in the caption of Fig. 1. In general, we now write explicitly through the paper which kind of source subtraction has been done in the images. We also added a column about the specific source subtraction to EDT1, following the referee's comment below.

>> The spectral index of the megahalos, although steeper than the hosted radio halo, are in line with those found for ultra-steep spectrum radio halos. There is no discussion as to whether these classes may be related, or be related to similar physical mechanisms.

A: We thank the referee for raising this point. Ultra steep spectrum radio halos are thought to be produced by Fermi-II acceleration under conditions that are less favorable than those in classical radio halos, i.e. less energetic merger events and consequently longer acceleration time scales. The spectrum of megahalos suggests long re-acceleration time scales, similar to those in ultra steep spectrum radio halos. We have thought about adding a comment about this in the paper, however, we fear that mentioning ultra steep spectrum radio halos would not be clear for non expert readers, unless we dedicate a paragraph to the introduction of these sources, which is challenging because of the word limit.

>> Related to the possibility of this being a new class, it would be useful to report the expected megahalo 1.4 GHz flux density range for the four clusters, and discuss this in relation to the power-mass correlation for the given mass range

A: We have compared the radio power of megahalos with the radio power-mass correlation found for radio halos by Cuciti et al (2021) at 1.4 GHz and by van Weeren et al. (2020) at 150 MHz. Depending on the assumed spectral index, megahalos would lie from a few times to ~100 times below the correlations.

These correlations however have been derived for sources with typical $(R_H/R_{500})^3$ (proportional to the volume) of 0.1 (see Fig. 3 in Cuciti et al. 2021), while megahalos are at least an order of magnitude larger, therefore their radio power cannot be directly compared to the radio power of radio halos. For this reason, it is more accurate to compare the emissivity of radio halos and megahalos, to take into account the large difference in terms of size.

>> In several places 'error' is used when 'uncertainty' is more appropriate – if error values are known, they can be corrected for.

A: We thank the referee for catching this. We replaced “error” with “uncertainty” when appropriate.

>> References: paper shows a strong preference for referencing co-author work. Although I acknowledge that several coauthors are leaders in the diffuse emission fields, more effort could be made to cite more broadly and acknowledge the work of other teams/individuals.

A. Following the advice of the Editor, we have substantially reduced the introduction in the main text. We have consequently removed some references, e.g. Murgia+ 2004, Stuardi+ 2021, Dolag+ 2001, Vazza +2014, van Weeren 2010, Cassano+13. We have added the following references: Iapichino+2011, Nelson+2014, Miniati15, Pinzke+17, Colafrancesco+2003.

>> Figure 2: suggest changing the colour of the 30arcsec resolution halo – currently it's the same as the lowest brightness of the megahalos, which may be aesthetically pleasing but can be confusing for image interpretation. A grey region may be appropriate instead.

A: Done

>> Figure 2: Nowhere is it discussed what the inset image is. After reading the full document, it's clear that this is the same image as shown in EDF3, however the image details should be given in the Fig 2 caption.

A: We added this sentence to the caption of the Figure: “On the right we show the 60" resolution 144 MHz image and the anuli we used to measure the surface brightness reported in the plot on the left (see Methods section for more details).”

>> line 108: “of maintaining”  “for maintaining”

A: Done.

>> line 122: “with what proposed”  “with what is proposed”

A: Done

>> lines 128-129: why would the macro/micro-physics change abruptly?

A: We thank the referee for pointing this out. We do not think that the properties of the plasma must change abruptly, but the observed differences between the radio halo and megahalo regions suggest that some properties of the plasma change moving towards larger distances from the cluster center. We have reworded the sentence.

>> Table 1: why are there different robust weighting iterations across clusters? How are the optimal weightings chosen?

A: In general, the robust weighting is chosen to have the best compromise between the rms noise of the image and the sensitivity to large scale emission. In particular, for the images that we used to subtract the sources, we tuned the robust weighting, in combination with the taper parameter, and we checked the model images until we found the value that allows the cleaning algorithm to best pick the clean component associated to the extended sources (radio galaxies and/or radio halo) that we want to subtract.

>> EDF1: Why are different sigma limits used for the two clusters (2sig for ZwCl0634 and 3sig for A665)? Sigma limits for interpretation should be uniformly chosen so as not to create bias.

A: We have changed the first contour of EDF1 (a) from 2 sigma to 3 sigma. We have modified the green region accordingly. The flux density measurements inside the green region have slightly changed, but the derived spectral index is consistent with the previous version.

>> EDF1: The red and green regions are difficult to see in the left panel, particularly the red one. Suggest changing the red region to magenta or cyan in both cases.

A: We have changed the color of the red regions to magenta.

>> line 320: Why are different angular scales chosen (45' vs 28')? Are these related to a physical scale at the cluster redshift, or is it arbitrary/linked to observed emission size?

A: The size of the regions is chosen manually to be as small as possible (in order to ensure faster calibration and imaging) but still include sufficient compact source flux density to obtain robust calibration solutions. This depends on the local radio environment around the cluster field. We clarified this in the text.

>> lines 326-327: "clearly visible at 20as – 30as" – I disagree, as this is not apparent in Fig 1 (black contours) except perhaps for ZwCl0634. If this is the wrong figure to be comparing to, tell the reader which figures show the "clearly visible" emission (EDF2 perhaps?).

A: We thank the referee for noticing this. In fact, the caption of Fig. 1 was wrong. We had not used the 3 sigma contours for all the images at 30" resolution in order to better highlight the different size of the radio halos. Now, we use 4 sigma for all the black contours in Fig. 1.

>> lines 348-349: "detected at best" – reword for better clarity, currently reads like the emission is not really visible but best seen at 2 arcmin resolution.

A: We reworded with: is detected at best -> is best detected in the...

>> EDF2: Suggest adding the R500 region to each panel so that the high resolution imaging can more readily be visually compared to the other low res images presented

A: Done.

>> EDT1: might be good to add a column with details about the relevant source subtraction, e.g., none, HR only, HR + RH. Related to an earlier comment, the phrase “source-subtracted” in the paper is ambiguous given the number of different subtractions used. Also might be useful to add a note for those images presented in other figures, to help the reader correlate the results between different parts of the paper, i.e., “the 30”, 60” and 2’ HBA images are presented in Figures XX, YY, and ZZ, respectively.”

A: We added two columns to EDT1, one with the details on the performed subtraction and one with the Figure(s) where the image is presented. We thank the referee for this suggestion that will help the reader connect different parts of the manuscript.

>> EDF3: right panels: note in the caption that these are after subtraction of the HR components only and include the RH emission.

A: Done.

>> EDF3: A665 left – the discontinuity is not as clear as for the other profiles and it is worth adding some discussion in the text as to why this may be

A: We agree with the referee that the discontinuity in Abell 665 is not as pronounced as for the other three profiles. Still, a second component, which is clearly not consistent with the radio halo profile, is present. In addition, the large-scale diffuse emission in Abell 665 shows differences with respect to the central halo also in terms of spectrum and emissivity, similarly to the other three clusters. Therefore we think it is correct to classify it as a megahalo. We have expanded the discussion about this topic in the “Surface brightness radial profiles” paragraph in the Methods section.

>> What is the bandwidth of the LOFAR 144 MHz data? Is it possible to obtain in-band spectral index maps for the megahalos? (if no, despite bandwidth availability, mention/discuss)

A: LoTSS observations are carried out in the frequency range 120-168 MHz. The narrow frequency range, combined with the uncertainties on the flux density scale of the in-band images, severely limits our ability to derive meaningful information from the in-band spectra, especially in the case of resolved low signal-to-noise sources such as megahalos (see section 3.7 in Shimwell et al. 2022 for an extensive discussion about in-band spectra in LoTSS images). We added a sentence to explain this in the paragraph “LOFAR HBA data reduction” in the Methods section.

>> EDF4: add boundary line to denote “southern part”

A: Done

>> EDF4: is there any spatial correlation between where the radio halo lies and any changes in the spectral index map?

A: The spectral index map in EDF4 shows a flatter spectral index in the region of the radio halo (which roughly corresponds to the second contour in this image) and a steeper spectral

index in the outer region, in agreement with our measurements of the integrated spectral indices. In the northern part, patches of relatively flat spectral index are visible also in the outer region, however, this region has been crossed by a shock wave, therefore it is possible that the shock has re-energised the emitting particle, contrasting the steepening of the spectrum. For this reason we do not use the northern region in the measure of the integrated spectral index of the megahalo.

>> lines 395-396: “bottom panels” – those are for A2218, the A665 panels are in the second row

A: Thank you for noticing the mistake, we have corrected it.

>> EDT2: keep the same order as in all the other figures and tables in the paper (ZwCl, A665, A697, A2218)

A: Done.

>> line 403: “best of our possibilities”  “best of our abilities given current techniques”

A: Done.

>> Emissivity section: include the equation used to calculate the emissivity, rather than just including a paper reference.

A: We added the equation to calculate the emissivity.

Response to referee 2

The authors would like to thank the referee for the careful reading of the manuscript and the helpful suggestions that improved the quality and readability of the paper. We list below the authors' reply to each comment raised by the referee.

C. Data & Methodology (technical/scientific issues):

1) Sample selection: Fig. 3 is an important plot that shows a key result, namely that LOFAR is identifying just the tip of the iceberg of these “megahalo” sources. However, for this statement to hold, Fig. 3 needs to show a well-defined and well-constructed sample of clusters for which the authors have uniform data. This is in fact the case of the Planck clusters in the LOFAR data release 2, for which the authors understand very well the data limitations and quality.

However, by adding ZwCl 0634.1 to the sample, it breaks down the argument. I understand that the megahalo discovered in this object highlights a “proto-typical” example, but I believe that the paper could be just as strong, and even stronger if the authors limit themselves to the Planck sample. It would make the results statistically complete.

Along these lines, if the authors feel strongly on keeping ZwCl 0634.1 in their paper, and that they are aware that Abell 2255 has a similar radio megahalo (as stated in line 159), then they should include Abell 2255 in their paper. Also in what context was ZwCl 0634.1 observed with LOFAR and how was its megahalo discovered? This needs to be explained/clarified in more detail.

A: ZwCl 0634.1+4750 is a Planck cluster observed as part of LoTSS, therefore it is not part of the sample only because it is outside the region of the sky that has been arbitrarily chosen as the DR2 footprint. On the other hand, the observations of A2255 that show emission on large scales are much deeper than LoTSS, indeed, in the LoTSS image of A2255 a possible megahalo is not visible (Botteon+ 2020). For this reason, we removed the sentence on Abell 2255 and we rephrased the sentence in which we list the clusters where we found megahalos in order to make it clear that they are all Planck clusters and they all belong to a sample for which we have homogeneous data.

2) In the paper, the authors make a strong claim that radio megahalos must be a distinct phenomenon from radio halos. This is a very important claim, which I do not believe the paper justifies. The authors demonstrate well that the properties are different (i.e. lower emissivity, different emissivity profile, steeper spectrum), but this does not imply that they must be of different origin or an entirely new phenomenon. In other words, their observations do not (yet) demonstrate that the seed population or the acceleration process must be completely different.

If the authors want to make this claim, they must demonstrate it. For example, can they produce numerical simulations that support this claim? Or do they have theoretical arguments that predict the existence of a distinct class of extended radio structures (versus regular halos)? Couldn't radio megahalos simply be an older, more diffuse population of radio halos? In this case, this does not make them distinct necessarily.

Also, lines 106 to 109, how is turbulence operating in a different regime? This needs to be clarified and justified quantitatively. Similarly, in lines 127 to 139, these are important claims, but without simulations or calculations to support them.

Related to this point, according to Extended Data Figure 3, Abell 665 appears to have a profile similar to its radio halo, so the distinction may not be as clear as claimed. Overall, if the authors feel strongly that radio megahalos are indeed a distinct class of radio sources, they need to provide additional evidence.

A: We agree with the referee that current observations do not allow us to claim that the physical processes at the origin of megahalos are new or completely different from what has been observed so far. In fact, one possible mechanism that we propose is turbulence, which is the same powering radio halos. However, our observations show that megahalos have different properties with respect to radio halos, and therefore demonstrate that the phenomenology, intended as the observable event, must be different from that of radio halos.

Merging clusters in cosmological simulations show a significant excess of turbulence in the central regions (Mpc-scales) of galaxy clusters. This merger-driven turbulence is responsible for powering radio halos, which are observed in dynamically active clusters. On the other hand, megahalos extend much beyond this region and show different observable properties (emissivity, brightness profile, spectrum). Consequently, they are unlikely to be powered by the same turbulent component. Megahalos cannot simply be older radio halos because, in the absence of additional in situ acceleration processes, the lifetime of the electrons responsible for the observed radio emission, 100-300 Myr, is way too short compared to the time that is necessary for the relativistic plasma to spread on the observed spatial scales. To support our claim that megahalos may be probing a different component of turbulence, we produced a new numerical analysis showing that a baseline level of turbulence in the ICM extends through the entire cluster volume and it is less sensitive to the presence of recent mergers. Although the details of the physical mechanisms remain to be addressed, megahalos are likely connected to this broader turbulent component. We added a paragraph to the Methods section to explain how we implemented the numerical simulations and to show their results. We also added relevant references to the text.

Regarding Abell 665, we agree with the referee that the discontinuity in surface brightness profile is not as pronounced as for the other three clusters. Still, a second component, which is clearly not consistent with the radio halo profile, is present. In addition, the large scale diffuse emission in Abell 665 shows differences with respect to the central halo also in terms of spectrum and emissivity, similarly to the other three clusters. Therefore we think it is correct to classify it as a megahalo. We have added a discussion about this to the “Surface brightness radial profiles” paragraph in the Methods section.

3) Minor comment related to Figure 2: The authors need to add a section in the Supplement series on how the gas density (computer simulation) was obtained and selected to match that of ZwCl 0634 and why that particular orientation was chosen. The authors should also add a warning note in the caption specifying that the figure is just for illustrative purposes to illustrate the extent expected of the gas density versus the megahalo (given that the computer simulation was not specifically tuned for ZwCl 0634).

A: We added a note to the caption of Fig. 2 to clarify that the simulated gas density distribution is only for illustrative purposes. We have used the density map of one of the clusters of the simulation that is now discussed in the Methods section. The map has been

obtained in a standard procedure, by projecting the gas density of all cells along the line of sight, using one of the coordinate axis of the simulation to make the projection.

F. Suggested improvements:

1) Figure 2 (as well as lines 98 to 101) and Figure 3: if LOFAR is detecting just the tip of the iceberg, is there a possibility that the megahalos actually extend further beyond than what the authors are able to detect currently with LOFAR? According to Extended Data Figure 3, the profiles appear to converge, so one could fit a profile and determine the total extent of the halo (out to about $r=1.5$ Mpc according to Extended Data Figure 3). How does this compare to the distribution of relativistic particles in computer simulations? And will LOFAR 2.0 or SKA be able to detect this? I would invite the authors to add a few sentences on this.

A: The profiles shown in EDF3 show a downturn at a radius of around 1.5 Mpc, and could be fitted with a power law with a cutoff. This suggests that the megahalo would not be much larger even if detected with deeper observations. What will substantially change with the improved sensitivity of LOFAR 2.0 and SKA is the possibility to detect new megahalos even in less massive clusters. In fact, when we say that we are likely detecting only the tip of the iceberg we mean that megahalos may be common to a large number of clusters, rather than a peculiarity of these four systems. We have slightly rephrased that sentence to make it clearer. Also, deeper observations will be useful to produce images at higher resolution where megahalos are detected and thus study the details of their morphology. Whether or not the large-scale profile of the megahalo emission can be reproduced by the combination of large-scale turbulence and a diffuse distribution of low-energy relativistic electrons can only be investigated with dedicated simulations in the future.

2) Figure 3: I would suggest adding a line (perhaps $\beta=1$) for LOFAR 2.0 and SKA, to illustrate what will be accessible in the near future.

A: We added the line corresponding to LOFAR 2.0 LBA observations (assuming $\beta=1$) to Fig. 3. We note that LOFAR 2.0 observations will be conducted with the international stations and thus will reach a resolution of ~ 1 arcsec. This will ensure the possibility to accurately subtract the contaminating sources before producing images at lower resolution and increase the sensitivity to the large scale emission.

On the other hand, there is no agreement yet on the final design of the SKA-LOW, in particular on the length of the maximum baseline (see Table 9 in https://astronomers.skatelescope.org/wp-content/uploads/2017/04/SKA-TEL-SKO-0000751-01_Cost_Control_Project_Report-signed.pdf for possible scenarios depending on the financial budget). This will not only affect the sensitivity itself but also the ability to subtract the contaminating sources. For this reason, we think that a quantitative assessment of the ability of SKA to detect megahalos is not reliable yet and we would prefer not to add a line to Fig.3.

3) Extended Data Table 2: I would suggest adding a bit more information that would be useful to the community. In particular, a column indicating the emissivity of the radio halos (so that the reader can directly compare to the megahalo) and perhaps other properties such as extent of the halo versus megahalos, spectral index, etc. I leave the authors to decide, but I believe such a table would be beneficial for the reader.

A: We thank the referee for this suggestion that improved the readability of our results. We added to EDT2 information about the radius, the emissivity and the spectral index, both for radio halos and megahalos. We have slightly modified the text in the emissivity paragraph to introduce the measurements reported in the table.

In the previous version of the paper we made an order of magnitude comparison between the emissivity of megahalos and the average emissivity of radio halos in the literature. Now we make a more precise comparison between the emissivity of megahalos and the corresponding radio halos in the same clusters and we report the correct number in the text.

4) Figure 1: I suggest making the figures larger to better illustrate the halos (they appear small on printed paper). Perhaps two columns by two rows would better highlight them. I also suggest adding the names of each cluster in the figures to make it easier for the reader.

A: We made the figures larger and we added the names of the clusters.

5) Since the publication is intended for Nature, it would be worth emphasizing the implications of the results for the broader community (beyond the cluster community). This should be added at the end of the main document.

A: We added two sentences to the main text to emphasize the implications of this work for the broader community.

H. Clarity :

1) When reading the abstract, I had initially understood that the authors had discovered a large radio halo surrounding 4 clusters, and not large radio halos surrounding 4 individual clusters of galaxies. I would rephrase the sentence “we report the first detection.... halos.”

A: We have rephrased the sentence. Now it is: we report four cases of the detection of a larger sea of radio emitting plasma around radio halos.

2) “These faint sources are called...Megaparsec.”. The authors need to rephrase the sentence because radio halos are not the only sources – radio mini-halos and relics are also known to exist.

A: We have rephrased the sentence. Now it is: These faint sources include radio halos and typically extend over ~ 1 Megaparsec.

3) Sometimes you cite Zwcl 0634 and sometimes Zwcl0634 and sometimes ZwCl 0634. I would make everything uniform and verify the rest of the cluster names as well.

A: Thank you for noticing this. The names of the clusters are now uniform through the manuscript.

4) Line 94 to 96: use to correct spectral indexes determined in the supplement series (with uncertainties) and not the approximate values.

A: Done.

5) Figure 2: In the caption, the authors should also specify that the insert (right panel) is at lower resolution of 60" (compared to Figure 1). I could not understand why it looked different compared to the image overlaid on the computer simulation until reaching the supplement series.

A: We added this sentence to the caption of the Figure: "On the right we show the 60" resolution 144 MHz image and the anuli we used to measure the surface brightness reported in the plot on the left (see Methods section for more details)."

6) Line 50: specify the central regions of clusters in kpc.

A: Since the Editor suggested removing the introductory material at the beginning of the main text, that sentence has been removed.

7) Line 66: for the term distinct, justify that it is distinct or refer to later on (where the authors demonstrate the different properties).

A: We removed "but distinct" from that sentence. We think that it reads better now and in fact the differences between the radio halos and the rest of the emission are explained later on.

8) Line 350: how were these diffuse bright sources identified and do these include head-tail sources, etc. If so, please clarify.

A: We added a few lines to clarify which sources we have masked.

Reviewer Reports on the First Revision:

Referees' comments:

Referee #1 (Remarks to the Author):

I thank the authors for responding to each of the comments raised. I am happy that each has been adequately addressed given the space limits.

Referee #2 (Remarks to the Author):

The authors have responded in detail to all my comments and made the appropriate changes.

I have read the revised version of the paper in detail and I only have some minor suggestions/corrections left to address:

- 1) Title : volume  volumes
- 2) Abstract : « Here we report... »  « Here we report in four distinct cases the detection...”.
- 3) Line 62: define lower frequencies in MHz.
- 4) Line 68-72: The authors should add a note that the error bars are quite large, so the evidence is only tentative.
- 5) Line 100: specify that the broader turbulence may be related to the accretion onto the cluster.
- 6) Caption of Figure 2: “The central region, outlined in red...” Isn't it outlined in black?
- 7) Figure 3: This is related to my first report. The authors mention that adding a line to illustrate SKA is probably too difficult and uncertain, but could the authors show the line for Meerkat? Or ngVLA (although it might be also too uncertain)?
- 8) Caption of Extended Data Figure 1 and 2: define “high-resolution” in arcsens.
- 9) Sometimes the authors describe the HBA as HBA and other times as the LOFAR 144 MHz data. I would make it uniform everywhere (so either use LOFAR 144 MHz or HBA or HBA LOFAR 144 MHz).
- 10) Lines 480 to 487. This is really interesting because the authors are essentially predicting that cool core (i.e. more relaxed) clusters may also have these megahalos. I would make this statement as it is a prediction that can easily be tested with LOFAR 2.0.

Author Rebuttals to First Revision:

Response to referee 2

The authors would like to thank the referee for the careful reading of the revised version of the manuscript and the helpful further suggestions. We list below our response to each of the comments raised by the referee.

1) Title : volume  volumes

A: Done

2) Abstract : « Here we report... »  « Here we report in four distinct cases the detection...”.

A: Following the Editor’s suggestion, we have slightly modified that sentence, which is now “Here we report observations of four clusters...”.

3) Line 62: define lower frequencies in MHz.

A: Done

4) Line 68-72: The authors should add a note that the error bars are quite large, so the evidence is only tentative.

A: Done.

5) Line 100: specify that the broader turbulence may be related to the accretion onto the cluster.

A: We added the sentence “likely related to the accretion of matter onto the cluster”.

6) Caption of Figure 2: “The central region, outlined in red...” Isn’t it outlined in black?

A: We thank the referee for spotting this typo. We corrected it.

7) Figure 3: This is related to my first report. The authors mention that adding a line to illustrate SKA is probably too difficult and uncertain, but could the authors show the line for Meerkat? Or ngVLA (although it might be also too uncertain)?

A: We agree with the referee that estimating the capability of the ngVLA in detecting megahalos is still too premature. We calculated the expected sensitivity of Meerkat plus, considering a spectral index for megahalos of -1.6. We used Meerkat plus instead of Meerkat because it will have higher sensitivity and approximately two times higher resolution than Meerkat and therefore will not be limited by confusion (<https://indico.ict.inaf.it/event/1515/contributions/9089/attachments/4387/9071/MeerKATplus.pdf>). However, given the steep spectral index of megahalos, the line would be quite close to the LOFAR 144 MHz line, therefore we believe that it would not add much information to the plot and it would make it more difficult to read.

8) Caption of Extended Data Figure 1 and 2: define “high-resolution” in arcsecs.

A: Done.

9) Sometimes the authors describe the HBA as HBA and other times as the LOFAR 144 MHz data. I would make it uniform everywhere (so either use LOFAR 144 MHz or HBA or HBA LOFAR 144 MHz).

A: We now use 144 MHz everywhere. We maintained “HBA” only in the few cases where the word “HBA” is coupled with “LBA”.

10) Lines 480 to 487. This is really interesting because the authors are essentially predicting that cool core (i.e. more relaxed) clusters may also have these megahalos. I would make this statement as it is a prediction that can easily be tested with LOFAR 2.0.

A: We added the sentence “Future deeper observations such as those that will be made with LOFAR 2.0 will allow us to test this scenario.”